# Dietary Carbohydrate Constituents Related to Gut Dysbiosis and Health

**DOI:** 10.3390/microorganisms8030427

**Published:** 2020-03-18

**Authors:** Ye Seul Seo, Hye-Bin Lee, Yoonsook Kim, Ho-Young Park

**Affiliations:** Research Division of Food Functionality, Korea Food Research Institute, Jeollabuk-do 55365, Korea; Seo.Ye-seul@kfri.re.kr (Y.S.S.); 50023@kfri.re.kr (H.-B.L.); kimyus@kfri.re.kr (Y.K.)

**Keywords:** dietary fiber, gut, high-carbohydrate diet, metabolic disorder, microbiota, sweetener, starch

## Abstract

Recent studies report that microbiota in the human intestine play an important role in host health and that both long- and short-term diets influence gut microbiota. These findings have fueled interest in the potential of food to promote health by shaping the intestinal microbiota. Despite the fact that large populations in Asia consume high quantities of carbohydrates, such diets have been ignored in comparison to the attention received by Western diets containing high quantities of fat and animal protein. We gathered data that suggest an association between imbalanced high-carbohydrate intake and gut microbiota and host health. In this review, we identify not only the effect of total carbohydrates on the intestinal microbiota specifically and the health of their hosts in general, but also how specific types of carbohydrates influence both factors.

## 1. Introduction

Microorganisms such as bacteria, protozoa, viruses, and fungi constitute more than ten times the number of somatic cells in the human body [1]. Among all human organ systems, the gut contains the greatest number of microorganisms, and their abundance and genetic diversity have been the focus of much research. Intestinal bacteria, 60% of which belong to the phyla *Bacteroidetes* and *Firmicutes*, play a variety of roles in human health [2,3]. They ferment and convert indigestible food such as fiber to available forms, produce bioactive substances, and control a wide range of biological mechanisms such as those underlying the immune system, glucose and energy homeostasis, and anti-inflammatory processes [4,5,6,7]. Moreover, studies report that human intestinal microbiota are associated with metabolic syndrome as well as specific diseases, including type 2 diabetes, cardiovascular disease, inflammatory bowel disease (IBD), Alzheimer’s disease (AD), and cancer [8,9,10,11]. For example, butyrate and other short-chain fatty acids (SCFAs) produced by gut microbiota appear to engage in direct anti-inflammatory activity; furthermore, Ridaura et al. (2013) revealed that obesity can be initiated by gut microbiota via microbial transplantation in mice [11,12]. Recent studies also found that gut microorganisms produce considerable quantities of amyloids known to play critical roles in AD pathogenesis [13].

Among the various factors known to affect the levels and composition of intestinal microbiota, such as delivery methods, lifestyle, heredity, and stress, diet is thought to be the primary contributor [14]. A study in which researchers compared the gut microbiota of European children with that of rural African children revealed that children in industrialized Europe, mainly consuming animal protein and fat, have a higher abundance of *Bacteroides* and lower abundance of *Prevotella* in their gut microbiota [15]. Inversely, rural African children consuming an agrarian diet showed lower *Bacteroides* and higher *Prevotella* proportions [16]. Furthermore, studies have demonstrated that changes in diet can cause an acute yet temporary alteration of gut microbiota composition within 24 h [17]. Taken together, these observations suggest that human intestinal microbiota can be manipulated by alterations in diet to treat and prevent metabolic syndrome.

Although the vast majority of people living in Asia consume high carbohydrate diets that influence their gut microbiota, previous studies in this field have commonly focused on Western diets characterized by the consumption of high quantities of fat and animal protein. Thus, this review will explore how diets high in carbohydrates affect metabolic syndrome by shaping the gut microbiota.

## 2. High Carbohydrate Diets and Metabolic Disorders

Dietary carbohydrates are often the primary energy source for residents of unurbanized regions. Recent developments in trade have made it easier to consume more concentrated carbohydrates; consequently, metabolic syndrome rates, which seem to be associated with high carbohydrate intake, are on the rise [18,19]. In this review, we will explore how diets high in carbohydrates affect metabolic disorders (Table 1).

Dietary carbohydrates are largely categorized as digestible and nondigestible. Digestible carbohydrates are used to gain energy via degradation by digestive enzymes [7]. Starches, usually from grains and tubers, are the predominant energy source, especially in agricultural areas. Mono- and disaccharides, found naturally in fruit, are carbohydrates composed of one and two sugar molecules, respectively. They are used to add sweetness to foods. Nondigestible carbohydrates can be divided into fermentable and nonfermentable fibers. Fermentable fibers such as pectins, β-glucans, β-fructans, inulins, oligosaccharides, and some resistant starches are fermented by the intestinal microbiota, producing a variety of beneficial substances, including SCFAs [42,43]. The concept of prebiotics derived therefrom typically refers to “a selectively fermented ingredient that results in specific changes, in the composition and/or activity of the gastrointestinal microbiota, thus conferring benefit(s) upon host health” [44].

The effect of high carbohydrate diets on host metabolism is known to be dependent on the glycemic index (GI) and glycemic load (GL). The higher the GI and GL in diets, the more they increase the risk of metabolic disorders such as type 2 diabetes and cardiovascular disease [18]. High carbohydrate diets, particularly those that include large portions of processed grains with high GI and GL, result in conditions likely to exacerbate dyslipidemia, which is characterized by increased triglyceride (TG) levels and decreased high-density lipoprotein cholesterol (HDL-C) levels [45,46,47]. In addition, diets rich in foods with high GI decrease fatty acid beta-oxidation and increase adipogenesis and fat accumulation by downregulating carnitine palmitoyltransferase 1 (CPT-1) levels in mRNA [20,48]. Furthermore, high GL diets reduce leptin levels and increase energy storage and the level of inflammatory markers such as C-reactive protein [20,49,50]. Although the relationship was initially based on insufficient evidence from outdated epidemiological studies, recent discoveries from the meta-analysis of cohort studies clearly show an association between GI values and metabolic disorders such as hyperinsulinemia; in addition, up to an 8% higher risk of breast cancer was found when comparing the group with the lowest intake of high-GI foods to the group with the highest [51,52]. A study in Canada reported that dietary GI was associated with an increased risk of prostate cancer and that higher dietary GL significantly increased the risk of colorectal and pancreatic cancers [53]. Obesity, type 2 diabetes, and other metabolic disorders associated with high GI and GL diets result in brain damage caused by glycation, which is also associated with an increased risk of AD [54].

Recent studies suggest that insulin resistance can aggravate the adverse metabolic effects of high carbohydrate diets [18]. Data from the Shanghai Women’s Health Study (SWHS) showed a stronger association between rice intake, GI, and GL and diabetes risk in women with a higher waist-to-hip ratio and body mass index (BMI) than in those with lower values of these metrics [18,55]. Furthermore, a high intake of polished white rice does not appear to cause deleterious metabolic effects in lean and physically active individuals such as farmers, while it has become a significant risk factor for diabetes in urbanized Asian populations. These results indicated that a carbohydrate-rich diet induces alterations in the gut microbiota, which is strongly associated with obesity and insulin resistance in an animal study [56].

### 2.1. Starches

Chinese researchers assessing the association between starchy carbohydrate intake and metabolic syndrome suggested that the excessive consumption of carbohydrates from starchy foods was significantly associated with metabolic syndrome and hyperlipidemia, while carbohydrates from other food sources showed no association [20]. Diets composed primarily of starchy carbohydrates seem to be associated with increased visceral fat and serum TG levels, while they were negatively associated with HDL-C in certain studies [20]. Tragnone et al. (1995) reported that high starch intake is significantly associated with the incidence of Crohn’s disease and ulcerative colitis. In particular, refined grains and tubers with high GI and GL may induce the incidence of glycemia and increased insulin resistance. In contrast, whole grains seem to reduce the risk of metabolic syndrome, as the consumption of whole grains twice daily was associated with a 21% lower risk for diabetes [18,57,58].

### 2.2. Mono- and Disaccharides

Elevated consumption of sugar-sweetened beverages (SSBs) seems to promote the incidence of type 2 diabetes even after accounting for the effects of body weight [22]. High SSB intake manifesting high GL raises both blood sugar and insulin levels. It thereby may result in pancreatic β-cell exhaustion, which appears to be associated with an increased risk of type 2 diabetes and cardiovascular disease in the long term [23]. Sonestedt et al. (2012) revealed that a high intake of disaccharides, in particular, is associated with atherogenic lipoprotein phenotype. Disaccharides metabolized in preference to lipids in the liver cause increased hepatic de novo lipogenesis, dyslipidemia, and insulin resistance. They may also promote visceral adiposity. Some studies suggest that diets high in monosaccharides and disaccharides increase the risk of Crohn’s disease and ulcerative colitis as well as pancreatic cancer [26,27,59]. Conditions such as obesity and overweight accompanied by insulin resistance may also pose specific health risks [27].

Fructose, an integral sweetener of the food industry, is one of the key dietary catalysts in the development of metabolic disorders by inducing gut dysbiosis [60]. Fructose consumption has been shown to adversely affect the lipoprotein profile, leading to cardiovascular disease [24]. Moreover, high fructose intake increases visceral adiposity, lipid dysregulation, and insulin resistance, which are associated with increased risk for type 2 diabetes and cardiovascular disease [25].

### 2.3. Artificial Sweeteners

Artificial sweeteners were originally marketed as healthy alternatives to replace natural sugar. However, their consumption has been shown to interfere with established responses that contribute to glucose and energy homeostasis [29]. Moreover, artificial sweeteners not only stimulate adipogenesis and suppress lipolysis in a sweet taste receptor independent manner, but they are also more likely to cause glucose intolerance than pure glucose and sucrose [28,29]. Data from recent studies suggest a link between the consumption of artificially sweetened beverages and a variety of negative health outcomes, including increased risk of overweight and obesity, type 2 diabetes, metabolic syndrome, and cardiovascular events, particularly in adults [30,61]. Suez et al. discovered that noncaloric artificial sweeteners induce dysbiosis and metabolic abnormalities by altering the intestinal microbiota [29]. Accumulating evidence shows that artificial sweeteners are associated with an increased risk of hypertension, stroke, dementia, urinary tract tumors, and laryngeal cancer, while they are inversely associated with the risk of breast and ovarian cancers [32,33,34,61,62].

### 2.4. Nondigestible Carbohydrates

Both epidemiological and experimental studies suggest that dietary fiber is negatively associated with many metabolic diseases and conditions, including cardiovascular diseases, IBD, type 2 diabetes, and obesity [37,63,64,65,66]. A significant negative association between dietary fiber and colorectal cancer risk has been observed in Europe [38,39]. Furthermore, recent studies have demonstrated that a diet high in fiber is beneficial to patients with ulcerative colitis and Crohn’s, decreasing the incidence of these diseases [40,41,67]. In Europe, cereal fiber intake has been associated with a reduced risk of gastric cancer [38].

One mechanism by which dietary fiber is beneficial to bowel health is by increasing digesta mass. Incompletely fermented fibers, including insoluble nonstarch polysaccharides such as cellulose, increase digesta mass based on their presence alone, as well as by their ability to absorb water. An increase in digesta mass contributes to health by diluting toxins, reducing intracolonic pressure, and increasing the frequency of defecation [8]. Additionally, dietary fiber improves bowel health by stimulating fermentation, thereby resulting in bacterial proliferation [68]. Many of the health benefits of fiber are attributed to the effects of their fermentation by colonic microbes and their metabolites. Fiber is fermented to organic acids that act as an energy source for other bacteria, as well as for the intestinal epithelium and peripheral tissues [69]. SCFAs—the major end-products of carbohydrate fermentation—contribute to bowel health in various ways; for instance, they help to lower intracolonic pH, thereby inhibiting the growth and activity of pathogenic bacteria [8].

## 3. High Carbohydrate Intake and Intestinal Microbiota

Among the myriad factors known to affect the intestinal microbiota—including delivery methods, lifestyle, heredity, and stress—diet is known to be the primary contributor. Studies suggest that people harbor contrasting gut microbiota composition based on the predominant diets of their local villages [15,70]. Moreover, certain studies have demonstrated that changes in diet can cause an acute, albeit temporary, alteration in gut microbiota composition within 24 h [17]. Consequently, we sought to explore how diets high in carbohydrates affect intestinal microbiota (Table 2).

In a study comparing the intestinal microbiota of European children with that of rural African children in Burkina Faso (BF), rural children consuming agrarian diets harbored lower levels of *Bacteroides*, more abundant *Prevotella* and *Xylanibacter*, higher microbial richness and biodiversity, and increased levels of SCFAs compared to urban children in Italy. Notably, a much lower abundance of potentially pathogenic bacteria such as *Escherichia*, *Salmonella*, *Shigella*, and *Klebsiella* reside in the intestinal tracts of children in BF than in those of children in Europe [16]. Interestingly, gut microbiota adapted to the Western diet were also found in children in urbanized areas of BF in later studies, suggesting that the differences in gut microbiota between rural BF children and urban Italian children were not the result of racial differences. This pattern was also identified in another cross-cultural study by Yatsunenko et al. (2012), in which the gut microbiota of the residents of Amazonas in Venezuela, rural Malawi, and metropolitan areas of the US were compared, showing that carbohydrate-rich diets increased the proportion of *bactella* in gut microbiota composition compared to Western diets high in fat and animal proteins.

Recent studies on the mechanisms by which dietary mono- and disaccharides affect gut microbiota conclude that they tend to increase *Bifidobacteria* while decreasing *Bacteroides*. Human subjects’ diet with several polyols, including maltitol, lactitol, and isomalt, resulted in increased relative abundance of *Bifidobacteria* with reduced *Bacteroides* [71]. In another study, the addition of lactose to the diet resulted in these same bacterial shifts while also decreasing *Clostridia* species. It is noteworthy to consider that many species in *Clostridium* cluster XIVa are known to be associated with IBS [72]. Francavilla et al. reported that lactose supplementation also increases fecal concentrations of beneficial SCFAs [73]. These findings are interesting, as lactose is thought to be a potential bowel irritant. In contrast, artificial sweeteners seem to induce microbial alteration directly opposite to that induced by natural sugars as described above. For instance, saccharin-fed mice exhibited intestinal dysbiosis with an increased relative abundance of *Bacteroides* and reduced *Lactobacillus reuteri* [29]. *Bifidobacteria* is thought to lower the risk and/or symptoms of certain metabolic diseases, including IBD, colorectal cancers, and necrotizing enterocolitis in neonates; consequently, it is widely used as a probiotic [85].

In contrast to digestible carbohydrates, nondigestible carbohydrates are not enzymatically degraded in the small intestine. Instead, they reach the large intestine where they may undergo fermentation by resident microorganisms [7]. Dietary fiber is a superior source of microbiota-accessible carbohydrates that can be used by intestinal microbes for energy and by the host as a carbon source [13,86,87]. In the process, they modify the intestinal environment, thereby rendering them effective prebiotics [44]. Sources of prebiotics include whole grains high in fiber and nondigestible oligosaccharides such as fructan, polydextrose, inulin, fructooligosaccharide (FOS), galactooligosaccharide (GOS), and arabinooligosaccharide (AOS). High intake of these carbohydrates in 49 obese subjects resulted in an increase in microbiota gene richness, while a low intake of these substances has been shown to reduce total bacterial abundance [74,75]. One study reported that the consumption of GOS induced *Bifidobacterium* species that can effectively utilize GOS [76]. Administration of resistant starch—the other nondigestible carbohydrate—was observed to increase the abundance of *Bifidobacterium adolescentis*, *Ruminoccocus bromii*, *Eubacterium rectale*, and *Parabacteroides distasonis* [77,80]. Many studies report that a diet high in nondigestible carbohydrates consistently increases intestinal *Bifidobacteria* and lactic acid bacteria [78,81]. Additionally, polydextrose-, FOS-, and AOS-based prebiotics have been reported to decrease *Clostridium* and *Enterococcus* species [79,82,83]. High fiber intake also increases the microbial production of SCFAs, playing an important role in the immune system, protecting the colonic mucus barrier, and preventing IBD and colorectal cancer [88,89,90]. Accordingly, one study reported a reduced abundance of butyrate-producing bacteria in colorectal cancer patients compared to their levels in healthy people [91]. In addition, when excluding the influence of SCFAs, interactions between dietary fiber and gut microbiota are beneficial for host health; for example, they contribute to the release of ferulic acid, which has antioxidative and anti-inflammatory properties in plant cell walls, regulating the availability of nutrients [89,92].

Many studies reported that a high carbohydrate diet changes the intestinal microbiota, which is associated with metabolic diseases such as type 2 diabetes, cardiovascular disease, and IBD [29,71]. High sugar consumptions induced obesity, insulin resistance, inflammation, and metabolic dysfunction due to changes in the intestinal microbiota in animal studies [56,93]. There are still few studies on the correlation between microbiota changes due to carbohydrate diets and metabolic disorders in human trials, but it is now becoming increasingly clear with recent studies.

## 4. Conclusions

Recently, a multitude of studies have shown that the intestinal microbiota can intermediate between diet and metabolic syndrome while emphasizing its potential to alleviate prevalent metabolic disorders. This article reviewed the effect of high carbohydrate diets on metabolic syndrome and the intestinal microbiota mediating such disorders. Many studies suggest that high carbohydrate intake is associated with the expression of markers associated with metabolic disorders and diseases, including diabetes and cardiovascular disease. Interestingly, the link between diet and metabolic syndrome seems to be dependent on GI and GL in the diet. A high intake of starch and mono- and disaccharides induces an increased risk of metabolic syndrome, while artificial sweeteners—originally recognized as healthier alternatives—actually lead to a higher risk for metabolic syndrome than natural sugars. In contrast, accumulating evidence has shown that nondigestible carbohydrates, including dietary fiber and resistant starch, lower the incidence of metabolic syndrome and related diseases.

## Figures and Tables

**Table 1 microorganisms-08-00427-t001:** Metabolic disorders associated with carbohydrates.

Types of Carbohydrates	Details	Effects on Human Health	Ref.
Starches	Carbohydrates from starchy foods	Risk of metabolic disorders and hyperlipidemia ↑	[20]
	Carbohydrates from starchy foods	Visceral fat and serum TG level ↑	[20]
	Carbohydrates from starchy foods	HDL-C level ↓	[20]
	Total carbohydrate, starch and refined sugar	Risk of Crohn’s disease and ulcerative colitis ↑	[21]
	Refined grains and tubers	Risk of glycemia and insulin resistance ↑	[20]
Mono- and disaccharides	Sugar-sweetened beverages	Risk of type 2 diabetes ↑	[22]
	Disaccharides	Risk of cardiovascular disease ↑	[23]
	Fructose		[24]
	Fructose	Lipogenesis, dyslipidemia, and visceral adiposity ↑	[25]
	Fructose	Insulin resistance ↑	[25]
	Sugar	Risk of Crohn’s disease ↑	[21]
	Sugar Total sugars, sucrose	Risk of ulcerative colitis ↑	[21] [26]
	Total sugars, sucrose, fructose	Risk of pancreatic cancer ↑	[27]
Artificial sweeteners	Saccharin and acesulfame potassium	Adipogenesis ↑ and Lipolysis ↓	[28]
	Saccharin, sucralose, and aspartame	Insulin resistance ↑	[29]
		Risk of dysbiosis and metabolic abnormalities ↑	
	Diet soda	Risk of type 2 diabetes and cardiovascular disease ↑	[30]
	Artificially sweetened beverages	Risk of hypertension ↑	[31]
	Saccharin/cyclamate aspartame/acesulfame-K	Risk of urinary tract tumor and laryngeal cancer ↑	[32] [33]
	Saccharin/cyclamate aspartame/acesulfame-K	Risk of breast and ovarian cancer ↓	[32]
	Artificially sweetened beverages	Risk of stroke and dementia ↑	[34]
Nondigestible carbohydrates	Additional 10 g of dietary fibers/day	Risk of cardiovascular disease ↓	[35]
	Cereal fibers	Risk of type 2 diabetes ↓	[36]
	Total dietary fibers and cereal fibers		[37]
	Dietary fibers	Risk of colorectal cancer ↓	[38] [39]
	Dietary fibers (Particularly fruits)	Risk of Crohn’s disease ↓	[40]
	Wheat bran cereal		[41]
	Cereal fibers	Risk of gastric cancer ↓	[38]

**Table 2 microorganisms-08-00427-t002:** Alterations of gut microbiota associated with carbohydrates.

Types of Carbohydrates	Details	Effects on Gut Microbiota	Ref.
Long-term agrarian diets	High Starches and fibers	*Prevotella* and *Xylanibacter* ↑	[16] [15] [70]
		*Bacteroides* ↓	
		Microbial richness and biodiversity ↑	
		Quantities of potentially pathogenic strains ↓	
		Production of SCFAs ↑	
Mono- and disaccharides	Daily fruits	*Bifidobacteria* ↑	[71]
		*Bacteroides* ↓	
	Lactose	*Clostridia* ↓	[72]
		*Lactobacilli* ↑	
	Lactose	Production of SCFAs ↑	[73]
Artificial sweeteners	Saccharin	*Bifidobacteria* ↓	[29]
		*Bacteroides* ↑	
		*Clostridia* ↑	
		*Lactobacilli* ↓	
		Dysboisis	
Nondigestible carbohydrates	Fruits and vegetable fibers	Microbial richness and biodiversity ↑	[74]
	FODMAP ^1^		[75]
	FODMAP ^1^	*Bifidobacteria* ↑	[75]
	Galactooligosaccharides		[76]
	Resistant starch type 4		[77]
	Whole-grain cereals		[78]
	Fructo-oligosaccharide		[79]
	Resistant starch type 4	*Actinobacteria* ↑	[77]
		*Bacteroidetes* ↑	
		*Firmicutes* ↓	
	Resistant starch type 2	*Ruminococcus* ↑	[77]
	Resistant starches	*Ruminococcus* ↑ *Eubacteria* ↑	[80]
	Resistant starch type 2		[77]
	Resistant starches	*Eubacteria* ↑ *Parabacteroides* ↑	[80]
	Resistant starch type 4		[77]
	Whole-grain cereals	*Lactobacilli* ↑	[78,81]
	Oligosaccharides mixture ^2^	*Clostridia* ↓	[82]
	Polysaccharide peptides		[83]
	Polydextrose	*Enterococcus* ↓	[84]
	Polysaccharide peptides		[83]
	Fructo-oligosaccharide		[79]
	Long-term high fiber diet	Production of SCFAs ↑	[16]

^1^ Fermentable oligosaccharides, disaccharides, monosaccharides, and polyols. ^2^ Mixture of short-chain galactooligosaccharides, long-chain fructooligosaccharides, and pectin-hydrolysate-derived acidic oligosaccharides.

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
