# Peer review of "Dietary Carbohydrate Constituents Related to Gut Dysbiosis and Health"

_microorganisms, 2020, doi:10.3390/microorganisms8030427_

Round 1

Reviewer 1 Report

The review is very interested to read, it is consistent, well written as well as organize and the references are update.

Author Response

We highly appreciate the reviewer’s constructive and helpful comments on our manuscript. As suggested by the reviewer, we have carefully response (marked in blue) to address the reviewer’s comments and revised manuscript (marked in red). We hope that the reviewer will find our responses to the comments satisfactory.

Reviewer(s)' Comments to Author:

Reviewer 1.

The review is very interested to read, it is consistent, well written as well as organize and the references are update.

▶ We appreciated reviewer’s kind comment.

Reviewer 2 Report

Seo et al. summarized the effect of how specific types of carbohydrates influence the intestinal microbiota specifically and the health of individuals.

  1. It is well known that diets can influence gut microbiota. However, the relationship gut microbiota change and health need to be clarified. I suggest the authors address more about the linkage of change of gut microbiota and health. Especially, the risk increase (or decrease) of specific disease under indicated carbohydrate is mainly through dysbiosis, its related metabolites or indicated carbohydrate itself. For example, as the authors stat, high GI and GL food result in conditions likely to exacerbate dyslipidemia, decrease HDL-C levels, decrease fatty acid beta-oxidation and increase adipogenesis. What’s the contribution of gut microbiota to health in this condition?

There are many animal studies can provide some evidences, if no enough human data to answer this question. Maybe the gut microbiota change in certain carbohydrate is just by stand but relevant to health directly.

  1. Drinking beverages containing fructose is popular in modern people. I suggest if should be more addressed in text not just mention simply in Table 1.
  2. Page 6 lines 173-176 and lines 182-184. “Human subjects fed high levels of glucose, fructose, and sucrose in the form of dates harbored increased relative abundance of Bifidobacteria with reduced Bacteroides [69]”….. “Bifidobacteria is thought to lower the risk and/or symptoms of certain metabolic diseases including IBD, colorectal cancers, and necrotizing enterocolitis in neonates; consequently, it is widely used as a probiotic [83].” How do you interpret these descriptions? Do you suggest peoples take high levels of glucose, fructose, and sucrose to improve our microbiota ?

Author Response

We highly appreciate the reviewer’s constructive and helpful comments on our manuscript. As suggested by the reviewer, we have carefully response (marked in blue) to address the reviewer’s comments and revised manuscript (marked in red). We hope that the reviewer will find our responses to the comments satisfactory.

Reviewer(s)' Comments to Author:

Reviewer 2.

  1. It is well known that diets can influence gut microbiota. However, the relationship gut microbiota change and health need to be clarified. I suggest the authors address more about the linkage of change of gut microbiota and health. Especially, the risk increase (or decrease) of specific disease under indicated carbohydrate is mainly through dysbiosis, its related metabolites or indicated carbohydrate itself. For example, as the authors stat, high GI and GL food result in conditions likely to exacerbate dyslipidemia, decrease HDL-C levels, decrease fatty acid beta-oxidation and increase adipogenesis. What’s the contribution of gut microbiota to health in this condition?

There are many animal studies can provide some evidences, if no enough human data to answer this question. Maybe the gut microbiota change in certain carbohydrate is just by stand but relevant to health directly.

▶ The hypothesis of our study is that “diet affects health by mediating gut microflora.” In response to the reviewer’s comment, we revised the manuscript as below.

Line 92–94: These results indicated that carbohydrate-rich diet induces alterations in the gut microbiota, which is strongly associated with obesity and insulin resistance in the animal study [56].

Line 217–222: Many studies reported that high carbohydrate diet changes the intestinal microbiota, which are associated with metabolic diseases such as type 2 diabetes, cardiovascular disease, and IBD [29,70]. High sugar consumptions induced obesity, insulin resistance, inflammation, and metabolic dysfunction due to changes in the intestinal microbiota in animal studies [56,92]. There are still few studies on the correlation between microbiota changes due to carbohydrate diets and metabolic disorders in human trials, but it is now becoming increasingly clear with recent studies.

  1. Drinking beverages containing fructose is popular in modern people. I suggest if should be more addressed in text not just mention simply in Table 1.

▶ According to the reviewer’s comment, we insert sentences in the 2.2 section.

Line 118–122: Fructose, an integral sweetener of food industry, is one of the key dietary catalysts in the development of metabolic disorders by inducing gut dysbiosis [60]. Fructose consumption has been shown to adversely affect the lipoprotein profile, leading to cardiovascular disease [24]. Moreover, high fructose intake increases visceral adiposity, lipid dysregulation, and insulin resistance, which are associated with increased risk for type 2 diabetes and cardiovascular disease [25].

  1. Page 6 lines 173-176 and lines 182-184. “Human subjects fed high levels of glucose, fructose, and sucrose in the form of dates harbored increased relative abundance of Bifidobacteria with reduced Bacteroides [69]”….. “Bifidobacteria is thought to lower the risk and/or symptoms of certain metabolic diseases including IBD, colorectal cancers, and necrotizing enterocolitis in neonates; consequently, it is widely used as a probiotic [83].” How do you interpret these descriptions? Do you suggest peoples take high levels of glucose, fructose, and sucrose to improve our microbiota ?

▶ We thank the reviewer for the detailed comment and corrected the sentence as below:

Line 180–182: Human subjects diet with several polyols including maltitol, lactitol, and isomalt increased relative abundance of Bifidobacteria with reduced Bacteroides.

Round 2

Reviewer 2 Report

No other question.